# PoliFormer: Scaling On-Policy RL with Transformers Results in Masterful Navigators

**Kuo-Hao Zeng**     **Zichen Zhang**     **Kiana Ehsani**     **Rose Hendrix**     **Jordi Salvador**

**Alvaro Herrasti**     **Ross Girshick**     **Aniruddha Kembhavi**     **Luca Weihs**

PRIOR @ Allen Institute for AI
[poliformer.allen.ai](poliformer.allen.ai)

**Abstract:** We present POLIFORMER (**Poli**cy Trans**former**), an RGB-only indoor navigation agent trained end-to-end with reinforcement learning *at scale* that generalizes to the real-world without adaptation despite being trained purely in simulation. POLIFORMER uses a foundational vision transformer encoder with a causal transformer decoder enabling long-term memory and reasoning. It is trained for hundreds of millions of interactions across diverse environments, leveraging parallelized, multi-machine rollouts for efficient training with high throughput. POLIFORMER is a masterful navigator, producing state-of-the-art results across two distinct embodiments, the LoCoBot and Stretch RE-1 robots, and four navigation benchmarks. It breaks through the plateaus of previous work, achieving an unprecedented $85.5\%$ success rate in object goal navigation on the CHORES-$\mathbb{S}$ benchmark, a $28.5\%$ absolute improvement. POLIFORMER can also be trivially extended to a variety of downstream applications such as object tracking, multi-object navigation, and open-vocabulary navigation *with no finetuning*.

**Keywords:** Embodied Navigation, On-Policy RL, Transformer Policy

## 1 Introduction

Reinforcement Learning (RL) has been used extensively to train embodied robotic agents to complete a variety of indoor navigation tasks. Large-scale, on-policy, end-to-end RL training with DD-PPO [1] enables near-perfect *PointNav*[1] performance when using a shallow GRU-based [2] architecture. However, this approach fails to result in the same breakthroughs for harder navigation problems like Object Goal Navigation *(ObjectNav* [3]) where an agent must explore its environment to locate and navigate to an object of the requested type. RL approaches for ObjectNav have generally not advanced beyond shallow GRU architectures due to challenges presented by training instability and unreasonably long training times with wider and deeper models, such as scaled-up transformers [4].

In a departure from on-policy RL, which is sample inefficient and often uses complex reward shaping and auxiliary losses [5], Imitation Learning (IL) has recently shown promise for ObjectNav. Ehsani *et al.* (2023) [6] demonstrated that the transformer-based SPOC agent, when trained to imitate heuristic shortest-path planners, can be trained stably, is sample efficient, and is significantly more effective than prior RL approaches on their benchmark. SPOC, however, ultimately falls short of mastery, plateauing at a success rate of ~57%. Critically, as we show in our experiments, the performance of SPOC does not seem to improve significantly when further scaling up data and model depth; we suspect this is a consequence of insufficient state-space exploration as expert trajectory datasets frequently contain few examples of error recovery, which can lead to sub-optimal performance due to compounding errors [7] or non-trivial domain shifts during inference.

---

[1]Point Goal Navigation is the task of navigating to a goal location using privileged GPS coordinates.

8th Conference on Robot Learning (CoRL 2024), Munich, Germany.

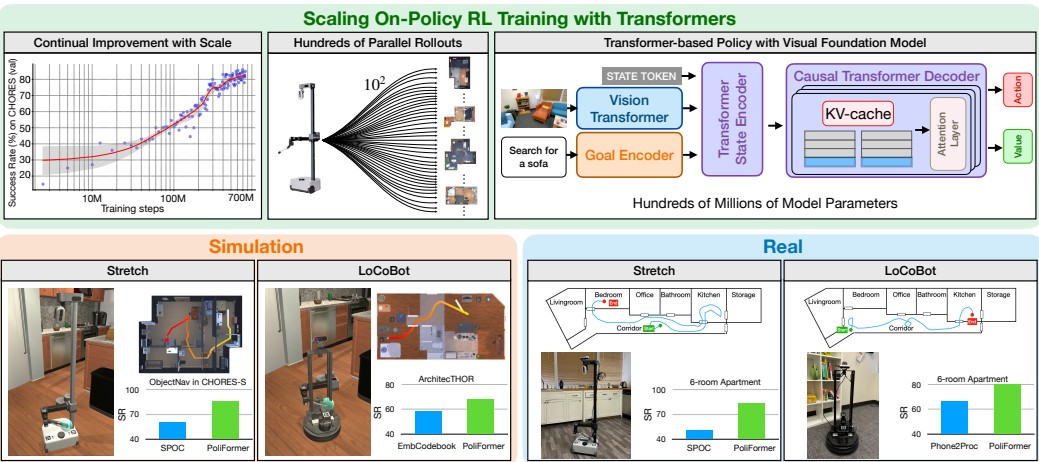

Figure 1: POLIFORMER, a transformer-based policy trained using RL at scale in simulation, achieves significant performance improvements in simulation (*bottom-left*) and the real world (*bottom-right*), across two embodiments. SR denotes Success Rate. We scale on-policy RL training across multiple dimensions: (*top-left*) we observe continual performance improvement with scaling RL training; (*top-middle*) we leverage hundreds of parallel rollouts for higher throughput; (*top-right*) we develop a transformer-based policy scaling model parameters to hundreds of millions.

In contrast to IL, RL requires that agents learn via interactive trial-and-error, allowing for deep exploration of the state space. This exploration has the potential to produce agents that can learn behaviors that are superior to those in the expert demonstrations. This raises the question: can we bring together the modeling insights from SPOC-like architectures and the exploratory power of RL to train a masterful navigation agent? Unfortunately, this cannot be done naively due to RL's sample inefficiency and complexities in using deep (transformer) architectures with RL algorithms. In this work, we develop an effective method for training large transformer-based architectures with RL, breaking through the plateaus of past work, and achieving SoTA results across four benchmarks.

Our method, POLIFORMER, is a transformer-based model trained with on-policy RL in the AI2-THOR [8] simulator at scale, that can effectively navigate in the real world without any adaptation. We highlight three primary design decisions that make this result possible. *(i) Scale in Architecture:* Building on the SPOC architecture, we develop a fully transformer-based policy model that uses a powerful visual foundation model (DINOv2 [9], a vision transformer), incorporates a transformer state encoder for improved state summarization, and employs a transformer decoder for explicit temporal memory modeling (Fig. 1, *top-right*). Importantly, our transformer decoder is causal and uses a KV-cache [10], which allows us to avoid huge computational costs during rollout collection and makes RL training affordable. *(ii) Scale in Rollouts:* We leverage hundreds of parallel rollouts and large batch sizes, which leads to high training throughput and allows us to train using a huge number of environment interactions (Fig. 1, *top-middle*). *(iii) Scale in Diverse Environment Interactions:* Training POLIFORMER with RL at scale in 150k procedurally generated PROCTHOR houses [11] using optimized Objaverse [12, 13, 14] assets results in steady validation set gains (Fig. 1, *top-left*).

POLIFORMER achieves excellent results across multiple navigation benchmarks in simulation. On CHORES-S, it achieves an impressive 85.5% Success Rate, a higher than +28.5% absolute improvement over the previous SoTA model [6]. Similarly, it also obtains SoTA Success Rates on Proc-THOR (+8.7%), ArchitecTHOR (+10.0%) and AI2-iTHOR (+6.9%). These results hold across two embodiments, LoCoBot [15] and Stretch RE-1 [16], with distinct action spaces (Fig. 1, *bottom-left*). In the real world (Fig. 1, *bottom-right*), it outperforms ObjectNav baselines in the sim-to-real zero-shot transfer setting using LoCoBot (+13.3%) and Stretch RE-1 (+33.3%).[2]

We further train POLIFORMER-BOXNAV that accepts a bounding box (*e.g.*, from an off-the-shelf open-vocab object detector [18, 19] or VLMs [20, 21]) as its goal specification in place of a given

---

[2]POLIFORMER even outperforms Phone2Proc [17], a baseline finetuned in 3D-reconstructed test scenes.

category. This abstraction makes POLIFORMER-BOXNAV a general purpose navigator that can be "prompted" by an external model akin to the design of Segment Anything [22]. It is extremely effective at exploring its environment and, once it observes a bounding box, beelines towards it. POLIFORMER-BOXNAV is a leap towards training a foundation model for navigation; with no further training, this general navigator can be used in the real world for multiple downstream tasks such as open vocabulary ObjectNav, multi-target ObjectNav, human following, and object tracking.

In summary, our contributions include: (i) POLIFORMER, a transformer-based policy trained by RL at scale in simulation that achieves SoTA results over four benchmarks in simulation and in the real world across two different embodiments. (ii) A training recipe that enables effective end-to-end policy learning with large-scale neural models via on-policy RL. (iii) A general purpose navigator, POLIFORMER-BOXNAV, that can be used zero-shot for multiple downstream navigation tasks.

## 2 Related Work

**IL and RL on Embodied Navigation.** Recent advancements include Point Goal Navigation [1], Object Goal Navigation [3, 23, 24, 25], Exploration [26, 27, 28], and Social Navigation [29], as well as successful sim-to-real transfer [6] and high performance in various downstream applications [16, 30, 31, 32, 33, 34, 35]. The prevalent approaches to building capable navigation agents include both end-to-end training [1, 6] and modular methods that leverage mapping [36, 37, 38] or off-the-shelf foundation models [39, 40, 41, 42, 43]. In these frameworks, the policy models are typically optimized via Imitation Learning (IL) using expert trajectories [44, 45, 46, 47, 48, 49, 50] or Reinforcement Learning (RL) within interactive environments [29, 51, 52, 53, 54, 55, 56, 57, 58, 59] and with carefully tuned reward shaping and auxiliary losses [5, 60, 61, 62, 63].

**Transformer-based Policies for Embodied Agents.** Recent works use transformer-based architectures for embodied agents. Decision Transformer (DT [64]) learns a policy offline, conditioning on previous states, actions, and future returns, providing a path for using transformers as sequential decision-making models. ODT [65] builds on DT and proposes to blend offline pretraining and online finetuning, showing competitive results in D4RL [66]. More recently, MADT [67] shows few-shot online RL finetuning on the offline trained DT in multi-agent game environments. The Skill Transformer [68] learns a policy via IL for mobile manipulation tasks. While the focus of these works is control towards specific tasks (relying on IL pre-training), we train *from scratch* using pure RL targeting a navigation policy that can be used zero-shot for many downstream tasks.

There has been a huge effort on improving the scale and training stability of transformer policies. GTrXL [69] proposes to augment causal transformer policies with gating layers towards stable RL training in [70]. PDiT [71] proposes to interleave perception and decision transformer-based blocks showing its effectiveness in fully-observed environments. GATO [72] learns a large-scale transformer-based policy on a diverse set of tasks, including locomotion and manipulation. Performer-MPC [73] proposes learnable Model Predictive Control policies with low-rank attention transformers (Performer [74]) as learnable cost functions. SLAP [75] learns a policy for mobile manipulation based on a hybrid design. Radosavovic *et al.* [76] learn a causal transformer using RL and IL for real-world humanoid locomotion. However, their input is proprioceptive state, without visual observations, and their learned policy is intended to be stationary (walking steadily in the real-world). NavFormer [77] learns image-goal conditioned transformer policies using offline-RL for target navigation. In contrast to existing approaches, we achieve efficient large-scale transformer-based on-policy RL training for a partially-observed, long-horizon, navigation task using RGB sensor inputs and show that our policy can be seamlessly deployed in the real world. We accomplish this without complex inductive biases or multi-task dataset aggregation due to the sheer amount of procedural scene layouts and visual variety of objects in simulation.

**Toward Navigation Foundation Models.** Many recent works attempt to produce general-purpose "foundational" navigation models. The causal transformer for humanoid outdoor locomotion [76], relying on proprioceptive state, shows emergent behaviors like arm swinging and in-context adaptation. GNM [78] trains image goal-conditioned policies (*i.e.*, conditioning on an image at the desired goal location) by IL across various embodiments, which allows them to scale up the size of their

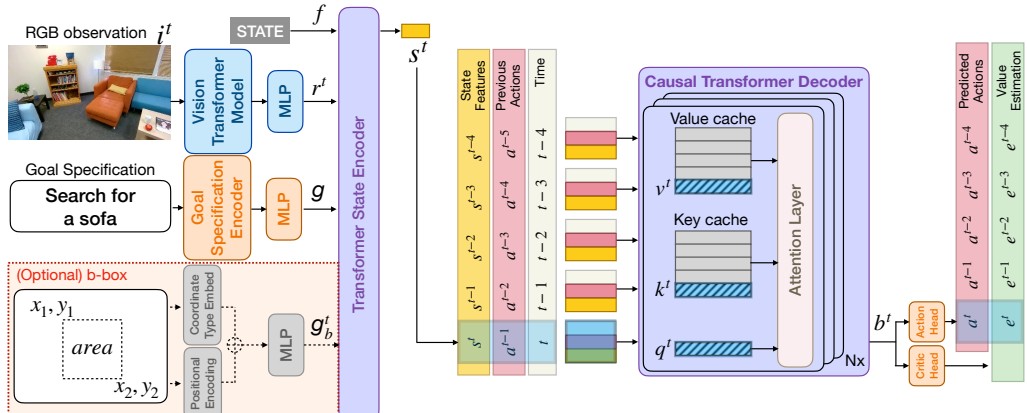

Figure 2: POLIFORMER is a fully transformer-based policy model. At each timestep $t$, it takes an ego-centric RGB observation $i^t$, extracts visual representations $r^t$ using a vision transformer model, further encodes state features $s^t$ using the visual representations and goal features $g$ (and optional detected bounding box goal features $g_b^t$), models state belief $b^t$ over time, employing a causal transformer decoder, and, finally, predicts action logits $a^t$ and a value estimation $e^t$ via linear actor and critic heads, respectively. For rollout collection and inference, we leverage the KV-cache [10] as our temporal cache strategy to prevent recomputing the forward pass for all prior timesteps at each new timestep, saving memory and speeding up both training and inference.

training data and manages to generalize to controlling unseen embodiments. NOMAD [79], which extends ViNT [80] uses a diffusion policy and also uses image goal conditioning. Unlike these works, we navigate from rich RGB image inputs and specify goals with natural-language text (or with bounding boxes), as such goals are far more easily available in real-world settings.

## 3 Method

RL, while effective and intuitive for training policies, has not yet been scaled in embodied AI to the same degree as in other domains. RL has primarily been applied to short-horizon tasks [81, 82], smaller environments [83, 84], or tasks utilizing privileged knowledge [1]. We scale learning in three directions; (i) Network Capacity, via POLIFORMER's deep, high-capacity, model architecture (Sec. 3.1); (ii) Parallelism and Batch size via our highly-parallelized training methodology that leverages large batch sizes for efficiency (Sec. 3.2), and (iii) Training Environments via our optimization of the simulation environment to support high-speed training (Sec. 3.3).

### 3.1 The POLIFORMER Architecture

We now detail POLIFORMER's transformer architecture (see Fig. 2), which is inspired by SPOC [6]. While we make several subtle, but important, changes to the SPOC architecture (detailed below), our largest architectural difference is in the transformer decoder: we replace the standard transformer decoder block [4, 85] with the implementation used in the Llama 2 LLM [86]. This change has dramatic implications for training and inference speed.

At each timestep $t$, POLIFORMER takes an ego-centric RGB observation $i^t$ as input and employs a frozen vision transformer model to extract a visual representation $r^t$. This representation, along with an embedding $g$ of the goal, are summarized by a transformer state encoder to produce the state representation $s^t$. The causal transformer decoder then encodes the state feature $s^t$ (along with $s^0, \ldots, s^{t-1}$) into a belief $b^t$. Finally, the linear actor and critic heads project $b^t$ to predict action logits $a^t$ and a value estimate $e^t$, respectively.

**Vision Transformer Model.** Inspired by prior work [6, 87], we choose DINOv2 [9] as our visual foundation backbone, because of its remarkable dense prediction and sim-to-real transfer abilities. The visual backbone takes an ego-centric RGB observation $i \in \mathbb{R}^{H \times W \times 3}$ as input and produces a patch-wise representation $r \in \mathbb{R}^{\frac{H}{14} \times \frac{W}{14} \times h}$, where $H$ and $W$ are the observation height and width, and $h$ is the hidden dimension of the visual representation. We reshape the visual representation into

a $\ell \times h$ matrix, $\ell = H \cdot W/196$, and project the representation to produce $v \in \mathbb{R}^{\ell \times d}$, where $d$ is the input dimension to the transformer state encoder. For effective sim-to-real transfer, it is important to keep the vision transformer model frozen when training in simulation.

**Goal Encoder.** For fair comparison, when training agents on ObjectNav benchmarks using the LoCoBot, we follow EmbCLIP [39] and use a one-hot embedding layer to encode the target object category. On benchmarks using the Stretch RE-1 robot, we follow SPOC [6] and use a FLAN-T5 small [88, 89] model to encode the given natural language goal and use the last hidden state from the T5 model as the goal embedding. Before passing the goal embedding to the transformer state encoder, we always project the embedding to the desired dimension $d$, resulting in $g \in \mathbb{R}^{1 \times d}$.

In select experiments (detailed in Sec. 4), we specify the goal object via a bounding box (b-box), either in addition to or as an alternative to text. In this case, the goal encoder processes the b-box using both the sinusoidal positional encoding and the coordinate-type embeddings to embed the top-left, bottom-right b-box coordinates and its area (5 values in total), followed by an MLP to project the hidden feature to the desired dimension $d$, resulting in $g_b \in \mathbb{R}^{5 \times d}$.

**Transformer State Encoder.** This module summarizes the state at each timestep as a vector $s \in \mathbb{R}^d$. The input to this encoder includes the visual representation $v$, the goal feature $g$ (and/or $g_b$), and an embedding $f$ of a STATE token. We concatenate these features together and feed them to a non-causal transformer encoder. This encoder then returns the output corresponding to the STATE token as the state feature vector. Since the transformer state encoder digests both visual and goal features, the produced state feature vector can also be seen as a goal-conditioned visual state representation.

**Causal Transformer Decoder.** We use a causal transformer decoder to perform explicit memory modeling over time. This can enable both long-horizon (*e.g.*, exhaustive exploration with back-tracking) and short-horizon (*e.g.*, navigating around an object) planning. Concretely, the causal transformer decoder constructs its state belief $b^t$ using the sequence of state features $\mathbf{s} = \{s^j|_{j=0}^{j=t}\}$ within the same trajectories.

Unlike RNN-based causal decoders [90] which, during rollout collection, require only constant time to compute the representation at each timestep, standard implementations of causal transformers require $t^2$ time to compute the representation at timestep $t$. This substantial computational cost makes large-scale on-policy RL with causal transformer models slow. To overcome this challenge, we leverage the KV-cache technique [10] to keep past feed-forward results in two cache matrices, one for **K**eys and one for **V**alues. With a KV-cache, our causal transformer decoder only performs feedforward computations with the most current state feature which results in computation time growing only linearly in $t$ rather than quadratically. While this is still mathematically slower than the constant time required for RNN-based decoders, empirically, we found only a small difference between KV-cache-equipped causal transformers and RNNs in overall training FPS for our setting. Compared to other temporal cache strategies, we found that KV-Cache offers the most significant speed improvements. Please see App. B.3 and Fig. 5 for a discussion of the impact on training speed resulting from different cache strategies.

### 3.2 Scalable RL Training Recipe

While KV-cache accelerates the causal transformer, this alone is not sufficient to enable efficient and fast RL training. In this section, we describe the methodology we use to achieve faster training. Although each of these individual findings may have been discussed in other works, we emphasize the critical importance of these hyperparameters for efficient training and, consequently, stellar results.

We parallelize the training process using multi-node training, increasing the number of parallel roll-outs by a factor of 4 compared to previous approaches [30, 59, 87] (we use 192 parallel rollouts for Stretch RE-1 agents and 384 for LoCoBot agents). We employ the DD-PPO [1] learning algorithm across 4 nodes, utilizing a total of 32 A6000 GPUs and 512 CPU cores. This scaling accelerates the training speed by approximately $3.4\times$, a near-linear gain compared to single-node training; we train for ~4.5 days with 4 nodes, while this would have required 15.3 days with a single node.

Our batch size during training, in number of frames, is equal to the number of parallel rollouts ($R$) multiplied by the length of these rollouts ($T$) and thus, by increasing $R$ as above, we multiplicatively increase the total batch size. We follow SPOC [6] and use a small *constant* learning rate of $2 \cdot 10^{-4}$ throughout the experiments. Instead of annealing the learning rate during training, we instead follow PROCTHOR [11] and *increase the batch size* by changing the rollout length from $T$=32, to $T$=64, and finally to $T$=128 (resulting in a final batch size of 49,152 frames for LoCoBot agents). We make these increases every 10M steps until we reach $T$=128.

### 3.3 Scaling Environment Interactions

For the experiments on LoCoBot, we use the original PROCTHOR-10k houses for training for a fair comparison to baselines. With the KV-cache technique, our training steps per second is ∼2.3k for training POLIFORMER, using 384 rollouts on 32 GPUs. For the experiments on Stretch, following [6], we use the PROCTHOR-150k houses with ∼40k annotated Objaverse 3D assets. As on-policy RL requires the agent to interact with the environment on-the-fly, we found the continuous physics simulation used by the Stretch RE-1 agent in AI2-THOR to be too slow to efficiently train at scale. To overcome this, we discretely approximate the agent's continuous movement via a teleportation-with-collision-checks approach. In particular, to move the agent, we perform a physics cast using a capsule collider representing the agent along the desired movement direction. If the cast does not hit any object (suggesting no potential collisions), we teleport the agent to the target location, otherwise we let AI2-THOR simulate continuous movement as usual. For rotations, we first teleport the agent to the target rotation pose. If the agent's capsule collides with any object, we teleport the agent back and let the AI2-THOR simulate the continuous rotation. For our tasks, these approximations increase simulation speed by ∼40%. We also found that AI2-THOR scene loading and resetting was a significant bottleneck when using Objaverse assets. We streamline AI2-THOR's Objaverse 3D asset loading pipeline by saving objects in the `Msgpack` format (natively usable with Unity's C# backend) as opposed to Python-only `Pickle` files. This change dramatically decreases the loading time of a new scene from ∼25s to ∼8s. With all these changes, our training steps per second increases from ∼550 to ∼950 using 192 rollouts on 32 GPUs for training Stretch RE-1 agent, reducing the training time by ∼42%.

## 4 Results

We now present our experimental results. In Sec. 4.1 we demonstrate that scaling RL with POLIFORMER produces SoTA results on four simulation benchmarks across two embodiments (LoCoBot and Stretch RE-1). Then, in Sec. 4.2, we show that POLIFORMER, despite being trained purely in simulation, transfers very effectively to the real world, outperforming previous work; we again show these results on the above two embodiments. In Sec. 4.3, we provide ablations for various design choices. Finally, we qualitatively show that POLIFORMER-BOXNAV can be extended to a variety of downstream applications in a zero-shot manner in Sec. 4.4.

**Baselines.** For our baselines, we chose a set of prior works in both imitation learning and reinforcement learning. SPOC [6] is a supervised imitation learning baseline trained on shortest path expert trajectories in AI2THOR. SPOC* is similar to SPOC, but is trained on more expert trajectories (2.3M *vs.* 100k). We build this baseline to verify if SPOC easily scales with more data. Emb-SigLIP [6] is an RL baseline also used by [6], but trained with comparable GPU hours with SPOC. The baselines [11, 63, 87] for LoCoBot-embodiment are all trained by RL using the same training steps used by POLIFORMER. We have three experiment configurations for our studies, specifying the goal as (a) natural language instruction text, (b) a b-box for the target object, and (c) b-box & text. Following [6], b-boxes in simulation are the *ground-truth* b-boxes, and so such models are not comparable fairly with RGB-only models. In the real world we replace GT b-boxes with outputs from the open-vocabulary object detector Detic [18], which uses only RGB images as input, and so all comparisons are fair.

**Implementation Details.** For LoCoBot benchmarks, we follow [87] to train our model and baselines on 10k training scenes in ProcTHOR houses for 435M training steps. The evaluation sets consist of 800 tasks in 20 scenes for AI2-iTHOR, 1200 tasks in 5 scenes for ARCHITECTHOR,

| Inputs | Model | Loss | CHORES-S ObjectNav Success (SEL) |
|---|---|---|---|
| RGB+text | SPOC [6] | IL | 57.0 (46.2) |
| | SPOC* | IL | 60.0 (30.5) |
| | EmbSigLIP [6] | RL | 36.5 (24.5) |
| | POLIFORMER | RL | **85.5 (61.2)** |
| RGB +text+b-box | SPOC | IL | 85.0 (61.4) |
| | POLIFORMER | RL | **95.5 (71.4)** |
| RGB+b-box | POLIFORMER | RL | 92.0 (73.9) |

(a) Stretch RE-1 on CHORES-S

| Inputs | Model | PROCTHOR-10k | ARCHITECTHOR | AI2-iTHOR |
|---|---|---|---|---|
| | | Success (SPL) | | |
| RGB+text | PROCTHOR [11][3] | 67.7 (49.0) | 55.8 (38.3) | 70.0 (57.1) |
| | SGC [63] | 70.8 (48.6) | 53.8 (34.8) | 71.4 (59.3) |
| | EmbCodebook [87] | 73.7 (48.4) | 58.3 (35.6) | 78.4 (23.7) |
| | POLIFORMER | **82.4 (58.5)** | **68.3 (45.1)** | 85.3 (72.7) |
| RGB +text+b-box | POLIFORMER | 90.4 (66.6) | 81.9 (55.6) | 94.9 (83.5) |
| RGB+b-box | POLIFORMER | 87.4 (56.2) | 85.7 (47.6) | 92.1 (78.6) |

(b) LoCoBot on ProcTHOR-10k (val), ArchitecTHOR and AI2-iTHOR (test)

Table 1: Across four ObjectNav benchmarks, POLIFORMER obtains SoTA performance. (a) Results on the CHORES-S ObjectNav benchmark, which uses the Stretch RE-1 embodiment, POLIFORMER dramatically outperforms the previous SoTA, IL-trained SPOC. (b) On three LoCoBot-embodiment test suites, POLIFORMER outperforms all prior work (all trained using RL).

| Model | | | PROCTHOR-10k (val) | ARCHITECTHOR |
|---|---|---|---|---|
| Vision Backbone | Encoder | Decoder | Success | |
| CLIP (ResNet50) | 1x CNN | 1x GRU | 67.7 | 55.8 |
| DINOv2 (ViTs) | 1x CNN | 1x GRU | 73.1 | 60.8 |
| DINOv2 (ViTs) | 3x Tx | 3x GRU | 73.6 | 59.8 |
| DINOv2 (ViTs) | 3x Tx | 1x Tx | 77.2 | 59.1 |
| DINOv2 (ViTs) | 3x Tx | 3x Tx | 80.4 | 63.9 |
| DINOv2 (ViTb) | 3x Tx | 3x Tx | **82.4** | **68.3** |

(a) Ablations on design choices for scaling model capacity

| Model | Stretch RE-1 | LoCoBot |
|---|---|---|
| ProcTHOR [11] | - | 26.7 |
| Phone2Proc [17] | - | 66.7 |
| SPOC [6] | 50.0 | - |
| POLIFORMER (ours) | **83.3** | **80.0** |
| SPOC+Detic [6] | 83.3 | - |
| POLIFORMER +Detic (ours) | **88.9** | - |

(b) Real-world results - Success

Table 2: We present (a) ablation studies on design choices for scaling up model capacity; and (b) the real-world results, on two different embodiements.

and 1500 tasks in 150 scenes for PROCTHOR-10k. For Stretch-RE1, we train our model and baselines on 150k ProcTHOR houses populated with Objaverse assets processed by ObjaTHOR. Then, we evaluate on 200 tasks in 200 scenes as in [6]. Training takes 4.5 days on 32 GPUs. Please see App. B for more details about training, inference, and environment setups.

## 4.1 POLIFORMER Achieves SoTA on four Benchmarks

Table 1a shows that, on the CHORES-S benchmark, POLIFORMER achieves 28.5% absolute gain in success rate over the previous best model. EmbSigLIP baseline results are taken from [6]; we suspect that its poor performance is, in part, due to its training budget being limited to match the training budget of SPOC.

**POLIFORMER works across embodiments.** Table 1b shows that the remarkable performance of POLIFORMER is not limited to one embodiment. The LoCoBot and Stretch-RE1 robots have different body sizes, action spaces, and camera configurations. Nevertheless, POLIFORMER is able to achieve 8.7%, 10.0%, and 6.9% absolute gain over the best baseline on PROCTHOR-10k (val), ARCHITECTHOR, and AI2-iTHOR.

## 4.2 POLIFORMER Generalizes to the Real World

We evaluated POLIFORMER on two real-world benchmarks for two different embodiments to show its sim-to-real transfer capabilities. We used the same evaluation set of 15 tasks for LoCoBot (3 different starting poses with 5 different targets), and 18 tasks for Stretch-RE1 (3 different starting poses with 6 different goal specifications), similar to [17] and [6], respectively. We do not use any real-world finetuning, and the testing houses were not seen during training. In the text-only setting, POLIFORMER achieves a remarkable improvement of 33.3% and 13.3% over the baselines for Stretch-RE1 and LoCoBot respectively (Tab. 2b). POLIFORMER even outperforms Phone2Proc [17], a baseline that has access to privileged information about the layout of the environment for finetuning.

## 4.3 Ablation Studies

The recipe for building the effective RL navigation model is the end product of several design decisions guided by experimentation, the combination resulting in our SoTA performance. Table 2a, shows the process of scaling POLIFORMER. Row 1 vs. Row 2: Moving from the CLIP ResNet-50 visual encoder to the ViT-small DINOv2 improves the success rate by an average of 5.2%.

---

[3]We report ProcTHOR results from [87] as these are the most up-to-date for recent AI2-THOR versions.

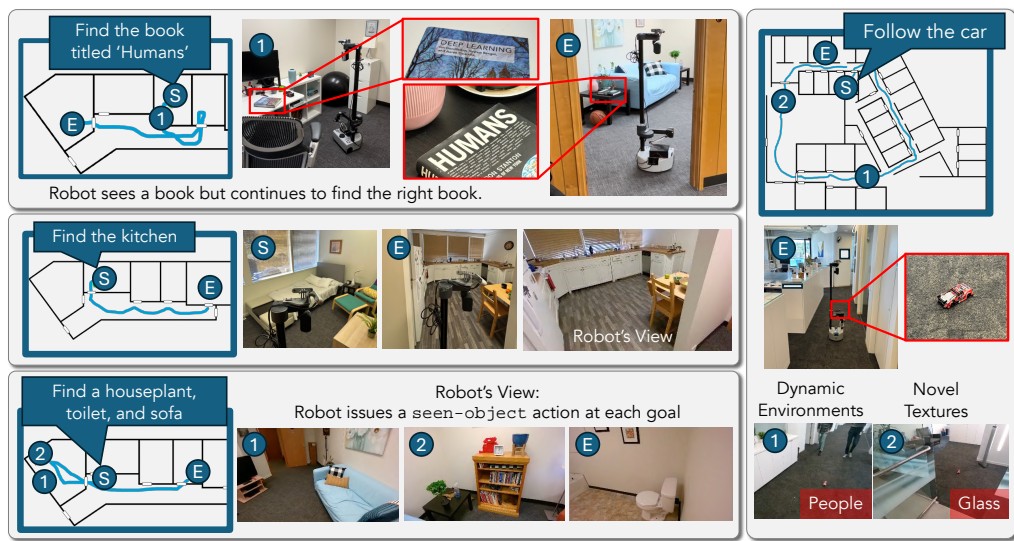

Figure 3: We use POLIFORMER-BOXNAV zero-shot to find a book with a particular title, navigate to a kitchen, navigate to multiple objects sequentially, and follow a toy car around an office building.

Row 3 *vs.* Row 5: Using a 3-layer transformer decoder instead of 3-layer GRU increases the performance by 5.5%. Row 4 *vs.* Row 5: Increasing the number of decoder layers from 1 to 3 increases the performance by 4%. Row 5 *vs.* Row 6: Using a larger capacity visual encoder (ViT-b instead of ViT-s), results in a 3.2% gain. In all, these changes result in a 13.6% absolute avg. improvement.

**Has POLIFORMER performance saturated?** Figure 1 (*top-left*) shows CHORES-S ObjectNav validation success rate as we train POLIFORMER for ~700M environment steps. As that plot shows, POLIFORMER's performance does not appear to have converged, and we expect that performance will continue to improve with more compute. This suggests that achieving near-perfect ObjectNav performance may only require further scaling our training approach.

### 4.4 Scaling POLIFORMER to Everyday Tasks

By specifying POLIFORMER's goal purely using b-boxes, we produce POLIFORMER-BOXNAV. While POLIFORMER-BOXNAV lacks the ability to leverage some helpful priors about where objects of certain types should reside (comparing rows 6 & 7 in Table 1a we see POLIFORMER-BOXNAV performs slightly worse than a model having both b-box and text goal specifications), this design decision makes POLIFORMER-BOXNAV a **fully general-purpose, promptable, navigation system** that can navigate to *any goal specifiable using a b-box*. Figure 3 shows four qualitative examples where, by using an open-vocabulary object detector (Detic [18]) and a VLM (GPT-4o [91]), POLIFORMER-BOXNAV is able to, zero-shot, navigate to (1) a book with a particular title, (2) a given room type, (3) multiple objects sequentially, and (4) a toy car as the car is driven around an office building. Simply by specifying goals as b-boxes and leveraging off-the-shelf systems, we obtain complex navigation behaviors that would otherwise need to be trained for individually: a painful process requiring new reward functions or data collection. Finally, we encourage readers to visit our website to view more qualitative results presented in video format.

## 5 Discussion

**Limitations:** Training RL agents for long-horizon tasks with a large search space requires extensive compute and demands careful reward shaping. While we believe POLIFORMER is capable of scaling to other tasks, it requires crafting new reward models for novel tasks such as manipulation. More discussion on limitations in App. E. **Conclusion:** In this paper we provide a recipe for scaling RL for long-horizon navigation tasks. Our model, POLIFORMER, achieves SoTA results on four simulation benchmarks and two real-world benchmarks across two different embodiments. We also show that POLIFORMER has remarkable potential for use in downstream everyday tasks.

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

## Appendices for *PoliFormer: Scaling On-Policy RL with Transformers Results in Masterful Navigators*

These appendices contain additional information about our:

- Zero-shot real-world applications (App. A),

- Training procedure (App. B),

- Environment, benchmarks, and quantitative real-world experiments (App. C),

- Simulation evaluations (App. D), and

- Limitations (App. E).

Please find our project website (see the poliformer.allen.ai) that contains

- Six real-world qualitative videos where PoliFormer performs the everyday tasks of Section 4.4 (recall also Figure 3), and

- Four qualitative videos in simulation showing our PoliFormer's behavior in the four benchmark environments (CHORES, ProcTHOR, AI2-iTHOR, and Architec-THOR).

## A   Details about Zero-shot Real-world Downstream Applications using an Open-Vocab Object Detector and VLM

By specifying PoliFormer's goal purely using b-boxes, we produce PoliFormer-BoxNav. PoliFormer-BoxNav is extremely effective at exploring its environment and, once it observes a bounding box, takes a direct and efficient path towards it. We now describe how we utilize this behavior to apply PoliFormer-BoxNav zero-shot to a variety of downstream applications by leveraging an open vocabulary object detector (Detic [18]) and a VLM (GPT-4o [91]).

**Open Vocabulary ObjectNav**. To perform open vocabulary object navigation (*i.e.*, where one must navigate to any given object type), we simply prompt the Detic object detector with the novel object type, for example, `Bicycle`. As PoliFormer-BoxNav relies on the b-box as its goal specification, it finds a bicycle in the scene smoothly.

**Multi-target ObjectNav**. To enable multi-target object goal navigation, we make a few simple modifications to the inputs and output of the Detic detector. On the input side, we query with multiple prompts simultaneously (one for each object type); for instance, `HousePlant`, `Toilet`, and `Sofa`, as shown in Fig. 3 (bottom-left). We then, on the output side, only return the b-box with the highest confidence score. Since the returned b-box also contains the predicted object type, we know what the target object the agent finds is when issuing a `Done` action. Therefore, we remove the found target from the list of target types, and reset the PoliFormer's KV-cache. If the agent issues a `Done` action without a detected b-box, we terminate the episode and consider it a failure. As a result, the agent is required to find all the targets from the list of target types to succeed in an episode.

**Human Following**. We change the Detic prompt to `Person`. Once a b-box is detected, PoliFormer drives the agent to approach it. Our experiment participant continues to walk away, so the agent keeps approaching them to minimize the distance.

**Object Tracking**. In this example, we control a remote control car that moves in the environment, and prompt the agent to find the car. Similar to **Human Following**, we change the prompt to `Toy Truck` in this example. As a result, the agent keeps trying to move closer to the detected b-box of the RC car, while avoiding collisions with objects in the dynamic scene.

**Room Navigation**. In this example, shown in Fig. 3 (middle-left), we provide no detections to the agent. As the agent sees no detections, it continuously explores the scene. As the agent explores, we

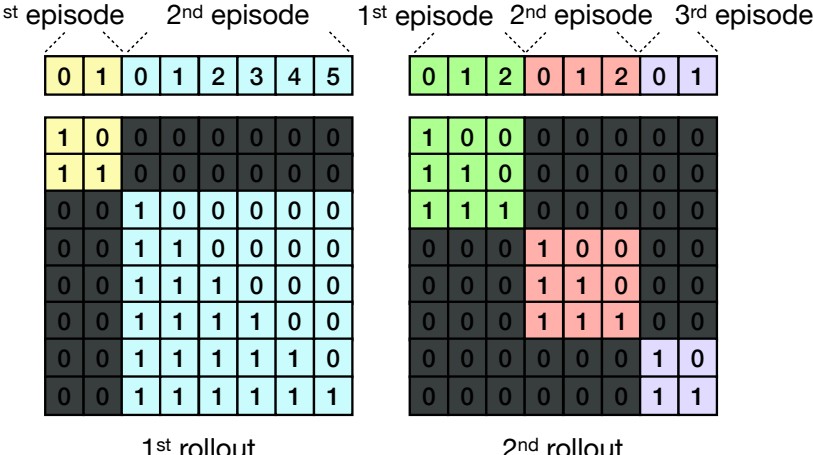

Figure 4: Attention Masks for training with block lower triangular structure.

query GPT4-o every 5 timesteps with the prompt `Am I in a Kitchen? Please return Yes or No.` with the most recent visual observation. Once GPT-4o returns `Yes`, the agent issues a `Done` action to end the episode.

**Instance Description Navigation**. In this example, shown in Fig. 3 (upper-left), the agent is prompted to find a specific book titled "Humans". Detic can generate open-vocabulary bounding boxes using instance-level descriptions but we found that doing this alone leads to high false-positive rates. To reduce these errors, we use GPT4-o to filter positive detections from Detic. In particular, a sample filtering prompt is "Is there a book titled "Humans" in this image? Please return Yes or No.". We find this combination works well in practice. The agent, not GPT-4o, remains responsible for deciding when it has successfully completed its task, and in the Fig. 3 example sees many books in its search but perseveres and eventually finds the correct one.

## B  Additional Training Details

### B.1  Reward Shaping

For reward shaping, we follow EmbCodebook [87] and PROCTHOR [11] and use the implementation in AllenAct [92]: $\mathcal{R}_{penalty} + \mathcal{R}_{success} + \mathcal{R}_{distance}$, where $\mathcal{R}_{penalty} = -0.01$ encourages an efficient navigation, $\mathcal{R}_{success} = 10$ when the agent successfully completes the task ($= 0$ otherwise), and $\mathcal{R}_{distance}$ is the change of L2 distances from target between two consecutive steps. Note that we only provide a nonzero $\mathcal{R}_{distance}$ if the new distance is less than previously seen in the episode. We do not enforce a negative reward for increasing distance. This formulation encourages exploration.

### B.2  Episodic Attention Mask

During training, to ensure that the causal transformer decoder cannot access observations or states across different episodes, we construct the episodic attention mask to only allow the past experiences within the same episode to be attended. In Fig. 4, we show a couple of possible rollouts collected during training. With the episodic attention mask, observations and states in an episode can only attend to previous ones within the same episode, in contrast with a naive causal mask where they could also potentially attend to observations and states in previous episodes.

### B.3  Different Temporal Cache Strategies

Besides the KV-Cache, in this subsection, we ablate four different temporal cache strategies (shown in Fig. 5 top):

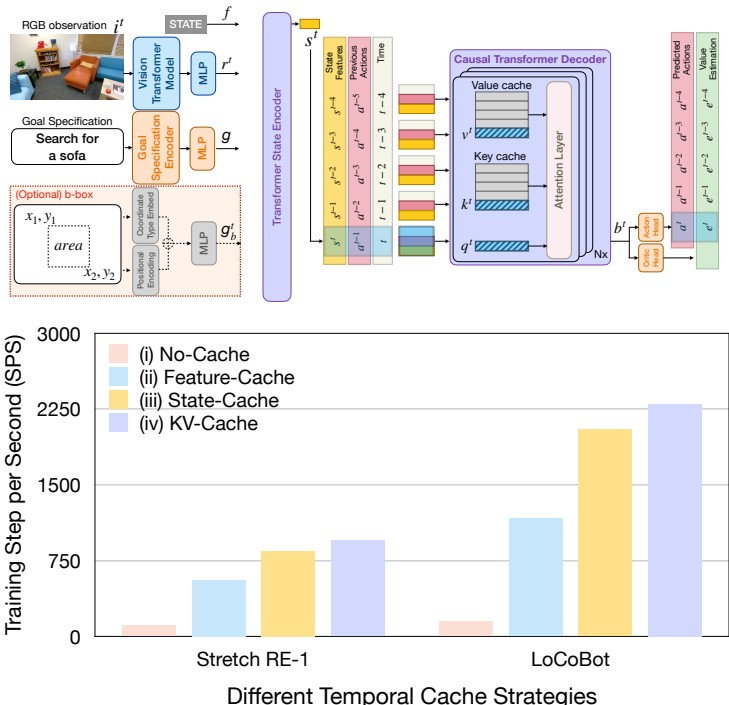

Figure 5: Different temporal cache strategies and their impact on the training speed. We ablate four different cache strategies, including (i) No-Cache, (ii) Feature-Cache, (iii) State-Cache, and (iv) KV-Cache, shown at top. The bottom chart shows the training Step per Second (SPS) achieved by different strategies, on both LoCoBot and Stretch RE-1 agents.

(i) No-Cache, we cache the raw frames (visual observation $i^t$) to provide past experiences for the causal transformer decoder. Therefore, POLIFORMER has to rerun the feedforward across all modules with the cached frames and the latest visual observation at each timestep.

(ii) Feature-Cache, we cache the features, including visual representation $v^t$ and goal embedding $g^t$ (as well as $g_b^t$). In this case, POLIFORMER needs to recompute the transformer state encoder and causal transformer decoder with the cached features and the latest feature at each timestep.

(iii) State-Cache, we cache the state feature vector $s^t$. Since it is right before the causal transformer decoder, POLIFORMER only requires to pass the cache state features and the latest state feature vector to the decoder, without recomputing the visual transformer model, goal encoder, and transformer state encoder.

(iv) KV-Cache, as described in Sec. 3.1, we cache the **K**eys and **V**alues inside the causal transformer decoder, further reducing the required computation time for the transformer decoder from $t^2$ to $t$ theoretically. In addition, since this strategy operates at very end, all the recomputations required by modules preceding causal transformer decoder can be saved.

The speed profile results are shown in Fig. 5 bottom. It clearly shows that placing the cache closer to the causal transformer decoder improves the training efficiency significantly.

## B.4 Hyperparameters for Training

Tab. 3 lists the hyperparameters used in our training and model architecture design. Please find more details such as scene texture randomization, visual observation augmentations, and goal specification randomization when using text instruction in our codebase.

| Training and Model Details | |
|---|---|
| **Parameter** | **Value** |
| Allowed Steps | 600 (Stretch RE-1), 500 (LoCoBot) |
| Total Rollouts | 192 (Stretch RE-1), 384 (LoCoBot) |
| Learing Rate | 0.002 |
| Mini Batch per Update | 1 |
| Update Repeats | 4 |
| Max Gradient Norm | 0.5 |
| Discount Value Factor $\gamma$ | 0.99 |
| GAE $\lambda$ | 0.95 |
| PPO Surrogate Objective Clipping | 0.1 |
| Value Loss Weight | 0.5 |
| Entropy Loss Weight | 0.01 |
| Training Stages | 3 |
| Steps for PPO Update  Stage 1 | 32 |
| Steps for PPO Update  Stage 2 | 64 |
| Steps for PPO Update  Stage 3 | 128 |
| Transformer State Encoder Layers | 3 |
| Transformer State Encoder Hidden Dims | 512 |
| Transformer State Encoder Heads | 8 |
| Causal Transformer Deocder Layers | 3 |
| Causal Transformer Deocder Hidden Dims | 512 |
| Causal Transformer Deocder Heads | 8 |

Table 3: Hyperparameters for training and model architecture.

## C   Additional Details about Environment, Benchmarks, and Real-World Experiments

**Action Space**. Following prior work using AI2-THOR, we discretize the action space for both LoCoBot and Stretch RE-1. For LoCoBot, we discretize the action space into 6 actions, including {MoveAhead, RotateRight, RotateLeft, LookUp, LookDown, Done}, where MoveAhead moves the agent forward by 0.2 meters, RotateRight rotates the agent clockwise by 30° around the yaw-axis, RotateLeft rotates the agent counter-clockwise by 30° around the yaw-axis, LookUp rotates agent's camera clockwise by 30° around the roll-axis, LookDown rotates agent's camera counter-clockwise by 30° around the roll-axis, and Done indicates that the agent found the target and ends an episode. We follow previous works [11, 17, 87] to use the same action space for LoCoBot for a fair comparison. For Stretch RE-1, we remove the LookUp and LookDown camera actions, and add MoveBack, RotateRightSmall, and RotateLeftSmall to the action space, where MoveBack moves the agent backward by 0.2 meters, RotateRightSmall rotates the agent clockwise by 6° around the yaw-axis, and RotateLeftSmall rotates the agent counter-clockwise by 6° around the yaw-axis. Again, this action space is identical to the one used in prior work [6] for fair comparison.

**Success Criteria**. We follow the definition of Object Goal Navigation defined in [3], where an agent must explore its environment to locate and navigate to an object of interest within an allowed number of steps $n$. The agent has to issue the Done action to indicate it found the target. The environment will then judge if the agent is within a distance $d$ from the target and if the target can be seen in the agent's view. An episode is also classified as failed if the agent runs more than $n$ steps without issuing any Done action. Across different benchmarks, $n$ and $d$ vary depending on the scenes size and complexity and agent's capabilities. We follow ProcTHOR [11] to use $n = 500$ and $d = 1$ meter for LoCoBot, and follow CHORES-$\mathbb{S}$ [6] to use $n = 600$ and $d = 2$ meters for Stretch RE-1.

**SPL and SEL**. Success Weighted by Path Length (SPL) and Success Weighted by Episode Legnth (SEL) are two popular evaluation metrics to evaluate how efficient an agent is to find the target. SPL is defined as $\frac{1}{N} \sum_{i=1}^{N} S_i \frac{l_i}{max(l_i, p_i)}$, where $N$ is the total number of episodes, $S_i$ is a binary

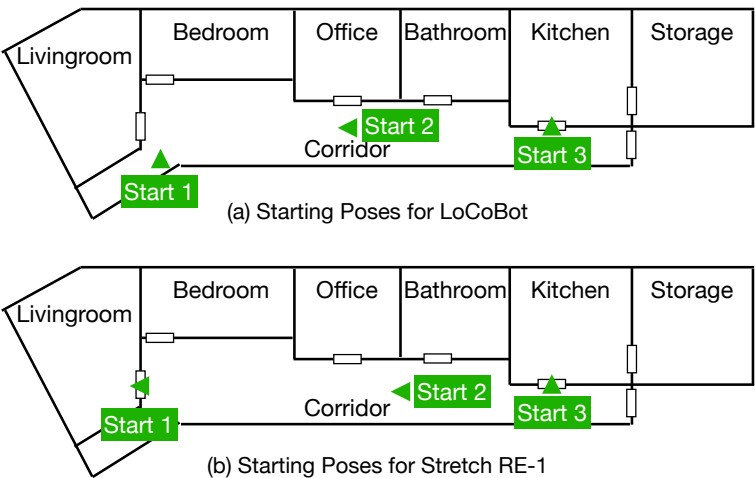

Figure 6: Starting Poses of (a) LoCoBot and (b) Stretch RE-1 used in the real world experiments. The arrow direction indicates where the agent faces with.

indicator of success for episode $i$, $l_i$ is the shortest travel distance to the target, and $p_i$ is the actual travel distance. SEL is defined similarly: $\frac{1}{N} \sum_{i=1}^{N} S_i \frac{w_i}{max(w_i, e_i)}$, where $w_i$ is the shortest number of steps to find the target, and $e_i$ is the actual number of steps used by the agent. By definition, SPL focuses on how far the agent has traveled, while SEL focuses on how many steps the agent has used (which also penalizes excessive in-place rotation). SPL can be derived by computing the geodesic distance between the agent's starting location and the target's location, while SEL needs a planner with privileged environment information to calculate the number of steps of expert trajectories. Therefore, we follow ProcTHOR [11] to report SPL to evaluate the LoCoBot agent, since those benchmarks do not provide planner, while we follow CHORES-$\mathbb{S}$ [6] to report SEL, since expert trajectories are available.

**Real-world Experiment Setup**. For the experiments using LoCoBot, we follow Phone2Proc [17] to use the same five target object categories, including `Apple`, `Bed`, `Sofa`, `Television`, and `Vase`, and the three starting poses, shown in Fig. 6 (a). Among those target categories, `Apple` can be found in the Living room and Kitchen, `Bed` can only be found in the Bedroom, `Sofa` and `Television` can only be found in the Living room, and `Vase` can be found in the Livingroom, Corridor, Office, and Kitchen. For the experiments using Stretch RE-1, we follow SPOC [6] to use the same six target object categories, including `Apple`, `Bed`, `Chair`, `HousePlant`, `Sofa`, and `Vase`, and the three starting poses, shown in Fig. 6 (b). Among the categories not mentioned above, `Chair` can be found in the Living room, Office, and Kitchen, and `HousePlant` can be found in the Living room, Office, Bathroom, and Kitchen. Note that we use the hardware design intorduced in SPOC [6] for Stretch RE-1. Instead of using the off-the-shelf camera equipped on the Stretch RE-1 (due to its narrow field of view), we use an Intel RealSense 455 fixed camera, which has a vertical field of view of $59°$ and a resolution of $1280 \times 720$. The camera is mounted facing forward but pointing downward, with the horizon at a nominal angle of $30°$. Please find more details in App. C.1. in SPOC [6].

**Details about Different Benchmarks**

First, during the training and testing stages, the possible target object types the agent is tasked with searching for include `AlarmClock`, `Apple`, `BasketBall`, `Bed`, `Bowl`, `Chair`, `GarbageCan`, `HousePlant`, `Laptop`, `Mug`, `Sofa`, `SprayBottle`, `Television`, `Toilet`, and `Vase`. We then outline the differences between the environments.

(i) AI2-iTHOR: This environment consists of 60 training scenes, 20 validation scenes, and 20 test scenes, and our training and evaluation never touch the training scenes. All the scenes

are created by human designers, including the structure, layout, and object placements, producing 750 episodes for validation and 800 episodes for testing. Moreover, iTHOR has 4 different styles of scenes, including LivingRoom, Kitchen, Bathroom, and Bedroom, where each style contains the objects which are semantically associated with it. Since each iTHOR scene only contains a single room, it doesn't require much exploration but rather focuses on recognition capability.

(ii) ArchitecTHOR: This environment consists of 10 high-quality large interactive houses designed by human designers, including 5 for validation and 5 for testing. Both validation and testing scenes have $1,200$ episodes each. It includes multiple rooms, larger navigable spaces, and more objects to explore. Since they are much larger than iTHOR scenes, they serve as a better benchmark to test the agent's exploration ability. Since ArchitecTHOR is designed by human designers, it also conforms to human priors on room layout and object placement.

(iii) ProcTHOR-val: This environment contains $1,550$ episodes across 150 procedurally generated houses. The way these houses are generated follows the same pipeline used in ProcTHOR. Therefore, the distribution of layout styles, number of rooms, and object placements respects the ProcTHOR-10k training houses' distribution, where our LoCoBot agent is trained.

(iv) CHORES-S: This environment contains 200 episodes across 200 procedurally generated houses, using the same approach as ProcTHOR. However, it includes the Objaverse 3D assets, presented by SPOC, which introduced $41,133$ assets into the ProcTHOR-150k houses, creating a more diverse scene distribution. Our Stretch Agent is trained on $80,000$ houses out of the ProcTHOR-150k training houses; the 200 CHORES-S test houses and episodes are not seen during training.

## D More Simulation Evaluations

**Performance Variance**. On CHORES-$\mathbb{S}$, since we follow SPOC [6] to apply test-time data augmentation and non-deterministic action sampling, we found that performance varies even using the same checkpoint, especially given that we are only evaluating on 200 episodes. As a result, we re-evaluate our POLIFORMER and SPOC*[4] 16 times and report mean success rate (mSR) and standard deviation (std). POLIFORMER achieves $82.5\%$ mSR with 1.897 std, while SPOC* achieves $56.7\%$ mSR with 2.697 std. This result indicates that POLIFORMER not only achieves a higher mSR than SPOC*, but also exhibits more reliably consistent behavior, *i.e.* a lower std, when run on the same episodes multiple times.

| Inputs | Model | Loss | **EasyObjectNav** | **RegularObjectNav** | **HardObjectNav** |
|---|---|---|---|---|---|
| | | | Success (SEL) | Success (SEL) | Success (SEL) |
| RGB+text | SPOC [6] | IL | 62.9 (40.5) | 48.2 (38.9) | 34.1 (27.4) |
| | SPOC* | IL | 69.7 (43.3) | 53.5 (34.3) | 31.0 (19.6) |
| | POLIFORMER | RL | **89.0 (62.1)** | **82.6 (71.8)** | **72.3 (62.8)** |
| RGB +text+b-box | SPOC | IL | 90.3 (67.7) | 78.7 (62.6) | 70.6 (52.5) |
| | POLIFORMER | RL | **98.1 (86.5)** | **90.4 (79.6)** | **86.0 (75.0)** |
| RGB+b-box | POLIFORMER | RL | 97.1 (83.2) | 91.9 (79.8) | 87.6 (75.0) |

Table 4: Large-scale evaluation results with different difficulty tiers. We evaluate performance on 2,000 episodes per tier.

**Larger Scale Simulation Benchmark using Stretch RE-1**. To further analyze POLIFORMER's performance through different difficulty settings, we construct 3 different levels of Object Goal Navigation benchmarks, `EasyObjectNav`, `RegularObjectNav`, and `HardObjectNav`, where each

---

[4]SPOC* is similar to SPOC but is trained on more expert trajectories (2.3M *vs.* 100k).

level contains 2k episodes, using Stretch RE-1. We construct these differentiated tasks by ensuring the oracle expert path length between the agent and target is 1 to 3 meters long for `EasyObjectNav`, greater than 3 meters for `RegularObjectNav`, and larger than 10 meters for `HardObjectNav`. The results are shown in Tab. 4. We observe that every model performs better as the agent is closer to the target at the episode start. In addition, on `EasyObjectNav` the agent barely needs exploration to find the target. Thereby, we find that POLIFORMER lagging behind POLIFORMER-BOXNAV by $\sim 9\%$ could result from a *Recognition Issue*. Moreover, the gap on `HardObjectNav` is widened to $\sim 13.7\%$, and it could result from an additional *Exploration Issue*. The performance gap between `HardObjectNav` and `EasyObjectNav` could also support that an *Exploration Issue* exists, but not just the *Recognition Issue*.

| Average Collision Rate (%) | PoliFormer w. Text goal | PoliFormer w. Box goal | PoliFormer w. Box + Text goal |
|---|---|---|---|
| ProcTHOR-val | 2.2 | 2.9 | 2.8 |
| ArchitecTHOR | 3.0 | 3.6 | 3.7 |
| AI2-iTHOR | 2.9 | 3.5 | 3.2 |

Table 5: Average Collision Rates for Different Benchmarks.

**Average Collision Rate**. As shown in Tab. 5. We measure the average collision rate achieved by POLIFORMER using the LoCoBot agent in ProcTHOR-val, ArchitecTHOR, and AI2-iTHOR. We compute the collision rate by $\frac{\#collision}{\#steps}$ within an episode and we average the rate across all evaluation episodes to obtain an average collision rate.

| ProcTHOR-10k val | PoliFormer w. Text goal | PoliFormer w. Box goal | PoliFormer w. Box + Text goal |
|---|---|---|---|
| x=0 | 82.4 (58.5) | 87.4 (56.2) | 90.4 (66.6) |
| x=1 | 82.2 (56.7) | 85.1 (55.0) | 89.8 (65.3) |
| x=2 | 80.1 (52.9) | 82.7 (51.1) | 86.1 (59.7) |
| x=3 | 74.5 (41.9) | 75.4 (42.3) | 83.4 (50.1) |
| **ArchitecTHOR** | **PoliFormer w. Text goal** | **PoliFormer w. Box goal** | **PoliFormer w. Box + Text goal** |
| x=0 | 68.3 (45.1) | 85.7 (47.6) | 81.9 (55.6) |
| x=1 | 67.0 (43.5) | 85.1 (47.2) | 81.9 (53.3) |
| x=2 | 63.3 (39.5) | 78.9 (43.6) | 78.2 (50.0) |
| x=3 | 60.1 (31.8) | 69.6 (34.5) | 71.5 (39.9) |
| **AI2-iTHOR** | **PoliFormer w. Text goal** | **PoliFormer w. Box goal** | **PoliFormer w. Box + Text goal** |
| x=0 | 85.3 (72.7) | 92.1 (78.6) | 94.9 (83.5) |
| x=1 | 85.3 (71.0) | 91.6 (78.3) | 93.4 (81.4) |
| x=2 | 85.1 (67.0) | 90.0 (74.1) | 92.6 (78.1) |
| x=3 | 80.8 (59.1) | 86.0 (64.3) | 90.6 (69.6) |

Table 6: Success Rate and SPL achieved by POLIFORMER with different level of noise across different benchmarks. We inject the gaussian noise $G(0, x * \sigma)$ into the action, where $x$ is a variable to control the scale of the noise, and $\sigma = \frac{\text{step size}}{3}$ for movement and $\sigma = \frac{\text{rotation degree}}{3}$ for rotation. The LoCoBot agent has a step size of 0.25m and rotation degree of 30 degrees by default.

**Robustness of POLIFORMER**. To evaluate the robustness of PoliFormer, we inject the gaussian noise into the action only at the evaluation stage. The results using LoCoBot across ProcTHOR-val, ArchitecTHOR, and AI2-iTHOR are shown in Tab. 6. POLIFORMER is robust to action noise even when experiencing substantial perturbations ($3 \times \sigma$ is a very large and unrealistic amount of noise).

# E   Additional Discussion on Limitations

**Depth Sensor**. It is important to note that POLIFORMER is not equipped with a depth sensor (which has been proven to be effective for manipulation). While the lack of the depth sensor does not affect our agent's performance on navigation, we acknowledge that integrating the depth sensor into our visual representation is an interesting direction for future work, especially when considering mobile-manipulation extensions.

**Discretized Action Space**. To have a fair comparison with baselines, we use the same discretized action space in this work (see App. C). The discretized action space might not be efficient and realistic in many real-world scenarios where the agent must act in a timely manner.

**Cross-embodiment**. In this paper, we demonstrate that we can train POLIFORMER using LoCoBot and Stretch RE-1. However, we have not yet explored training a single POLIFORMER for both embodiments. We leave this interesting research direction as future work.

**Further Scaling**. Our training and validation curves strongly suggest that even further scaling of model parameters and training time may lead to even more masterful models than those we have trained in this work. This perspective is exciting and we hope to enable further scaling with more computation resources and better visual foundation models in the near future.

**Failure Analysis**. The main mode of failure for POLIFORMER is the agent's limited memory. POLIFORMER clearly demonstrates memorization capabilities and is able to perform long-horizon tasks by exploring large indoor scenes without access to explicit mapping. However, as the trajectories get longer (specifically after visiting more than 4 rooms in an environment), the agent's recollection of the rooms it has explored deteriorates and the robot might re-visit rooms that it has explored previously.

**Extensive Use of GPUs and Significant Training Time**. We acknowledge that PoliFormer requires a large number of GPUs to train efficiently. Our main focus in this work is on pushing the performance of navigation agents to its limit without constraints on computing resources. Given the trend of GPU improvements over the last 10 years, we believe that the resources used in this work will become relatively commonplace in the near future. Empirically, we found that training with a smaller number of GPUs (e.g., 8 A6000 in a single host) could yield similar final performance, with the downside of slower training speed ($\sim 3.5\times$ longer to train compared to 32 A6000 in four nodes). Moreover, there are many interesting research directions in improving training speed/efficiency; for example, since our submission, we have found that removing the average pooling from PoliFormer's visual encoder can increase convergence speed by up to $2\times$. E.g., for the LoCoBot agent, we keep the number of tokens (e.g., 256 tokens) from DINOv2, instead of spatially average pooling them to $7 \times 7$ (resulting in 49 tokens). This implies that a better visual encoder with a larger receptive view over the visual observation could even further speed up training. Furthermore, we believe that incorporating more advanced techniques such as mixed precision training, flash attention, and advanced GPUs optimized for transformer models could reduce the training resource requirements.

**Sample Efficiency**. We hypothesize that an "IL pretraining +RL finetuning" paradigm could further improve sample efficiency. Early in on-policy RL training, policies trained from scratch are effectively random and thus produce a vast amount of marginally useful experiences. Pretraining with IL may improve sample efficiency by providing helpful priors on navigation and thus produce meaningful RL gradients faster. While it is nontrivial to implement this two-stage pipeline, we hypothesize that training PoliFormer using IL on expert data (e.g., the shortest path trajectories from SPOC) first and finetuning using on-policy RL could be an option to improve the sample efficiency. Since we focus on obtaining SoTA performance in this work, we believe that improving sample efficiency, by an IL+RL pipeline or with new RL algorithms such as SAPG [93], is an exciting research direction for future work.

