# OpenReview forum: "PoliFormer: Scaling On-Policy RL with Transformers Results in Masterful Navigators"
_robot-learning.org/CoRL/2024/Conference — CoRL 2024_

### Official Review · Reviewer_sGZ1 · 2024-07-19
**Impressive results, pushing SOTA, with questions on inference speed**

**Originality:** 4
**Technical Quality:** 5
**Clarity Of Presentation:** 5
**Potential Impact:** 4
**Recommendation:** 4
**Confidence:** 3

**Review:**

This is an excellent paper that is clearly written, understandable, and presents groundbreaking results advancing the state-of-the-art (SOTA).

Strengths:
* Training transformers in RL has been infeasible for a long time, and using a KV cache is an innovative approach to addressing the computational complexity problem for transformers.
* The clear improvement in SOTA results is remarkable.
* The real-world experiments with zero-shot sim-to-real transfer are particularly notable, especially given that the training environments are simulated. It is commendable that DinoV2 facilitates this zero-shot sim-to-real transfer.

Weaknesses:
* Despite the impressive real-world experiments, the fact that the experiments are sped up 20x raises questions. It would be beneficial to understand why the model operates at such a slow speed and what could be done to enhance inference speed.
* Training large models at this scale is commendable, but the extensive use of GPUs and significant training time could make reproducibility challenging for most research labs.
* There is no mention of releasing the code and/or trained models.

**Quality Of The Limitations Section:**

2

**Questions For Rebuttal:**

* L134: Reference to section within section should be fixed.
* Would linear transformers (state space models) be a viable alternative?
* Is a code-release planned? Are trained models planned to be released? This would be essential, especially since this work redefines SOTA.
* Please address limitations on inference speed, and on continuous control. What needs to be done to get a model that actually runs in real-time on the robot?

**Robotics Focus:**

4

**Summary Of Paper:**

This paper addresses the objectnav problem, where an embodied agent with a monocular camera navigates to an object specified by a bounding box. Unlike the simpler pointnav problem, which is considered solved with large-scale Reinforcement Learning (RL), objectnav benchmarks lag due to reliance on imitation learning, leading to stable but limited state space exploration. To overcome this, the authors propose training large transformers with RL. They use DinoV2 for image encoding, a transformer state encoder to combine goal specifications with image embeddings, and a transformer decoder for memorizing previous states, utilizing a KV cache for feasible large-scale RL training. The AI2-Thor environment generates diverse indoor scenarios, leading to state-of-the-art results across four benchmarks. The paper also suggests bounding boxes for goal specification, enhancing real-world applicability. Additionally, ablation studies and real-world experiments with two embodiments demonstrate successful zero-shot sim-to-real transfer.

**Summary Of Recommendation:**

This paper presents a compelling and well-executed approach to addressing the objectnav problem, leveraging large transformers and RL to achieve impressive results. Minor issues need to be addressed, such as improving inference speed and ensuring code and model release for reproducibility. Overall, the paper makes a significant contribution to the field.

---

### Official Review · Reviewer_jeoX · 2024-07-20
**Novel idea supported by strong experiments, good paper!**

**Originality:** 3
**Technical Quality:** 4
**Clarity Of Presentation:** 4
**Potential Impact:** 3
**Recommendation:** 4
**Confidence:** 3

**Review:**

**Quality:** Good. The proposed architecture and scaling recipe look reasonable and technically solid, and the experiment results look promising.

**Clarity:** Good. The structure and writing are good and easy to follow.

**Originality**: Good. The work with large-scale on-policy RL papers on robotics tasks is quite rare.

**Significance**: Good. This work can encourage more work in this direction to scale up on-policy RL on robot tasks.

**Strengths:**

1. RL with modern Transformers architectures is quite rare. That is partially because those architectures are more data-consuming, which is contrary to the long-perusing goal of the RL community to reduce the sample complexity. This work demonstrates that with large-scale training in simulation, RL agents can also achieve zero-short Sim2Real adaptation in navigation tasks.

2. The experiment results across four benchmarks and two robot embodiments show significant improvement in comparison with previous methods.

3. The ablation studies demonstrate the importance of design choices for the important components in architecture.

**Weakness**

1. As a reader who is not familiar with robot navigation, I feel confused about the differences and challenges of the selected benchmarks. It would be nicer to include a short description for each benchmark.

**Quality Of The Limitations Section:**

3

**Questions For Rebuttal:**

1. According to the description in Section 4, the agents were only trained in ProcTHOR environments and evaluated across four different benchmarks. Could the author explain a little bit about how different the training and evaluation environments are?

**Robotics Focus:**

4

**Summary Of Paper:**

This paper proposes an architecture and corresponding training recipe for end-to-end indoor navigation using reinforcement learning. The training process was fully conducted in simulation, and the rollouts in the real robots demonstrated its transferability into the real world. Experiments across two robot embodiments and four benchmarks show that the proposed method significantly outperformed baselines.

**Summary Of Recommendation:**

I highly recommend accepting this paper because of its novel idea and well-executed experiments.

---

### Official Review · Reviewer_Ax82 · 2024-07-22
**A solid advance in navigation using a flexible transformer-based method with a good, standardized evaluation.**

**Originality:** 4
**Technical Quality:** 4
**Clarity Of Presentation:** 4
**Potential Impact:** 4
**Recommendation:** 4
**Confidence:** 4

**Review:**

Transformer-based methods are increasingly used in robotic navigation problems. Building on a wide variety of prior work and ideas, the paper presents PoliFormer, which combines efficiency innovations in how the transformers are structured with large-scale training (both in policy rollouts enabled by efficiency, and on broad training in ProcThor, also empowered by work done to speed it up). Overall, the system shows a marked improvement on standardized navigation benchmarks. In addition, the paper presents a refactoring of the system which is slightly less performant, but serves as a foundation model for any task that can be specified with a bounding box. Nicely done.

Quality: Solid. Very good grasp of prior work, clear presentation of methods and advances, solid evaluations, and an innovative new structuring in PoliFormer-BoxNav.

Clarity: Good. Overall, the paper is well written and easy to follow. Section 3.1, however, appears to refer to itself in a few places (line 132, 134).

Originality: Good. The ideas are an advance over several existing lines of research, but a lot of work was put into each component to enable the increase in performance shown.

Significance: Solid. The paper has three key advances: a better training recipe (on top of a better architecture), an improvement to the state of the art, and PoliFormer-BoxNav as a new paradigm for formulating navigation foundation models.

**Quality Of The Limitations Section:**

3

**Questions For Rebuttal:**

The paper looks pretty good, though it might be worth checking it for minor typos, e.g.:
- Section 3.1, appears to refer to itself in a few places (line 132, 134).

**Robotics Focus:**

4

**Summary Of Paper:**

The paper presents a technical advance in training transformer-based architectures for navigation problems, evaluates it to show strong improvement on standardized baselines, and refactors it to provide a useful foundation model based on bounding boxes.

**Summary Of Recommendation:**

The paper includes a better method, an improvement to the state of the art, a new formulation of the nav problem, evaluation against standard benchmarks, and real-world tests.

---

### Official Review · Reviewer_bADd · 2024-07-23

**Originality:** 4
**Technical Quality:** 4
**Clarity Of Presentation:** 4
**Potential Impact:** 4
**Recommendation:** 4
**Confidence:** 5

**Review:**

## Quality
The paper exhibits a high level of technical quality, with a well-thought-out experimental design and thorough evaluation. The use of extensive training data and multiple benchmarks strengthens the validity of the results. The authors also provide detailed ablation studies to dissect the contributions of various components of PoliFormer.

## Clarity
Overall, the clarity of the paper is commendable. The authors present the complex concepts and methodologies in a structured manner. However, certain sections, such as the specifics of reward shaping and the precise hyperparameters used, could benefit from additional detail to aid in reproducibility and deeper understanding.

## Originality
PoliFormer’s integration of transformer architectures with on-policy RL is a novel contribution. This work pushes the boundaries of what has been achieved in embodied navigation tasks, particularly by leveraging the strengths of transformers in handling long-term dependencies and providing a robust framework for policy learning.

##Significance
The contributions of this work are significant in the field of embodied AI. By achieving state-of-the-art results and demonstrating robust sim-to-real transfer, PoliFormer sets a new standard for navigation tasks. Its ability to generalize to real-world scenarios without additional fine-tuning is particularly impactful, suggesting broader applicability and practical relevance.

##Strengths
- The combination of vision transformer encoders and causal transformer decoders represents a significant advancement in leveraging transformer architectures for RL tasks.
- PoliFormer achieves substantial improvements over existing models across multiple benchmarks, demonstrating robust performance.
- The model’s ability to perform well in real-world environments without additional fine-tuning highlights its robustness and practical applicability.

## Weaknesses
- On-policy RL’s inherent sample inefficiency requires extensive interactions, making it less practical for applications with limited computational resources. The significant computational resources needed for training (e.g., 32 GPUs, multi-node setups) limit the method's accessibility and scalability.
- While successful in navigation tasks, the model's generalization to other embodied AI tasks, such as manipulation or multi-agent interactions, remains unexplored and could require substantial re-engineering.

**Quality Of The Limitations Section:**

2

**Questions For Rebuttal:**

- Consider discussing potential strategies to improve sample efficiency, such as exploring off-policy methods or hybrid approaches that combine the strengths of on-policy RL with more sample-efficient techniques.
- The paper lacks detail on the specifics of reward shaping, which is crucial for long-horizon tasks. Including more information on the reward functions used and any techniques implemented to ensure effective exploration, such as intrinsic motivation methods or curiosity-driven exploration, would enhance the clarity and reproducibility of your work.
- Transformers introduce significant memory and computational overhead. Discuss any model compression techniques you have considered, such as pruning, quantization, or knowledge distillation, to mitigate these issues. This would be particularly relevant for deployment on resource-constrained robots.
- The evaluation primarily focuses on success rates and SPL. Adding additional metrics, such as collision rates, error recovery, and robustness under noisy conditions, would provide a more comprehensive assessment of the model's performance and robustness. This would offer a more complete picture of the model's capabilities and limitations.
- The paper does not discuss "NavFormer: A Transformer Architecture for Robot Target-Driven Navigation in Unknown and Dynamic Environments," which covers similar topics. Please revise your paper to include this work and clearly differentiate your contributions from it. Highlighting the unique aspects and advancements of PoliFormer in comparison to NavFormer will strengthen your paper.

**Robotics Focus:**

4

**Summary Of Paper:**

The paper introduces PoliFormer, an innovative reinforcement learning (RL) model for indoor navigation tasks that leverages a vision transformer encoder and a causal transformer decoder for enhanced long-term memory and reasoning. Trained extensively in simulation, PoliFormer achieves state-of-the-art results in object goal navigation, significantly outperforming previous models. The model excels in both simulated and real-world environments, demonstrating strong sim-to-real transfer capabilities without additional fine-tuning. Key contributions include scalable on-policy RL training using hundreds of parallel rollouts, effective use of transformers for navigation, and robustness across diverse tasks and environments. The paper also highlights potential extensions of PoliFormer to various downstream applications such as multi-object navigation and object tracking.

**Summary Of Recommendation:**

I strongly recommend the acceptance of this paper. The authors present PoliFormer, an innovative approach leveraging transformer architectures for on-policy reinforcement learning in indoor navigation tasks. The work demonstrates significant advancements in both architectural design and performance, achieving state-of-the-art results across multiple benchmarks. The model's robustness, scalability, and strong sim-to-real transfer capabilities are particularly noteworthy. Despite minor areas for improvement, such as enhancing sample efficiency and detailing reward shaping strategies, the paper's contributions are substantial and impactful. The thorough experimental validation and potential for broad applicability make this work a valuable addition to the field of embodied AI.

---

### Author Rebuttal · Authors · 2024-08-08

We thank all reviewers for their time and effort in reviewing our submission and providing valuable feedback. We’ve revised our manuscript and appendix to include the requested discussions, typo corrections, and evaluation results. All newly added content is highlighted in blue.

---

### Decision · Program_Chairs · 2024-09-04

**Decision:**

Accept

**Comment:**

Strengths:
- Use of transformer encoder / decoder for scaling RL
- Well written, with demonstrations across four benchmarks and two embodiments
- Rare mix of transformers and RL, all strong accepts from reviewers

Weaknesses
- Reliance on on-policy RL makes method less sample efficient. Method is highly compute intensive making it less accessible.

Reviewers unianimously agree on a strong accept for the paper after rebuttal phase.